# Unilateral Reverse Pupillary Block Associated with Multiple Ciliary Body Cysts and Pseudoexfoliative Syndrome

**DOI:** 10.3390/diagnostics15060758

**Published:** 2025-03-18

**Authors:** Idoia Goñi Guarro, Mia Zorić Geber, Rašeljka Tadić, Renata Iveković, Zoran Vatavuk

**Affiliations:** 1Department of Ophthalmology, University Hospital Centre Sestre Milosrdnice, 10000 Zagreb, Croatia; raseljkas@gmail.com (R.T.); renata.ivekovic@kbcsm.hr (R.I.); zo.vatavuk@gmail.com (Z.V.); 2Oftalmología Médica y Quirúrgica (OMIQ) Research, 08036 Barcelona, Spain; 3Department of Medicine and Pharmacology, Vrije Universiteit Brussel (VUB), 1090 Brussels, Belgium; 4School of Medicine (UZSM), University of Zagreb, 10000 Zagreb, Croatia

**Keywords:** angle-closure glaucoma, ciliary body cyst, ultrasound biomicroscopy, reverse pupillary block, PEX

## Abstract

The aim of this study is to present interesting images of a clinical case of asymmetrical bilateral ciliary body cysts associated with pseudoexfoliative syndrome (PEX), leading to unilateral reverse pupillary block and subsequent secondary angle-closure glaucoma in a 64-year-old patient who presented with vision loss and redness, revealing angle-closure glaucoma in the left eye. Slit lamp examination showed an asymmetrical iris configuration between the eyes, with a normal appearance in the right eye and an inverted “volcano-shape” iris appearance, corresponding to reverse pupillary block, with pseudoexfoliation in the left eye. Multimodal imaging confirmed the presence of bilateral ciliary body cysts, which were unexpectedly identified in the right eye. The patient’s secondary angle-closure glaucoma in the left eye was likely due to anterior displacement of the iris from these cysts. Following an inadequate response to topical and systemic treatments, the patient underwent trabeculectomy in the left eye, successfully stabilizing the intraocular pressure (IOP) and leading to the resolution of the reverse pupillary block. This case underscores the importance of thorough ocular examination and multimodal imaging in diagnosing complex clinical presentations like secondary angle-closure glaucoma stemming from the combination of ciliary body cysts’ pressure towards the angle, the pseudoexfoliative material component, and the reverse pupillary block configuration. All of the findings provided critical diagnostic clues leading to the identification of the underlying pathology.

**Figure 1 diagnostics-15-00758-f001:**
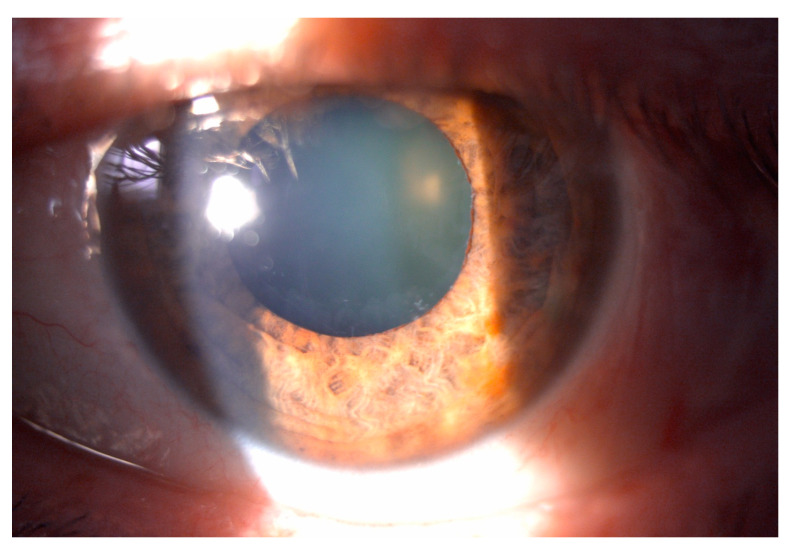
The Figure 1 shows a postoperative slit lamp biomicroscopy image of the left eye dilated, showing the pseudoexfoliative material in the anterior lens capsule and in the border of the iris. A 64-year-old man was referred to our Glaucoma Referral Center, Ministry of Health of the Republic of Croatia, complaining of red eye and visual loss in the left eye in recent months. He had no history of amblyopia or trauma in the left eye, no relevant medical history pathology and no reported allergies. Family ocular history was negative for glaucoma or other ocular pathologies. He had no symptoms in the right eye. His first registered IOPs were 14 mmHg in the right eye and 44 mmHg in the left eye. He was referred to our clinic under ocular therapy, which included a combination of a carbonic anhydrase inhibitor, a beta-blocker, and prostaglandin analogues, along with a systemic carbonic anhydrase enzyme inhibitor. His best corrected visual acuity (BCVA) was 20/20 in the right eye and 20/28 in the left eye, with a manifest refraction of +0.75 in the left eye. In our examination, his IOP, as measured by applanation tonometry, was 17 mm Hg in the right eye and 50 mmHg in the left eye. Pachymetry demonstrated central corneal thicknesses of 555 µm in the right eye and 537 µm in the left eye, and the axial length was 23.04 mm in the right eye and 22.83 mm in the left eye. The posterior segment examination showed a normal appearance of the optic nerve in the right eye, compared to a more paled appearance in the left eye. In the right eye, slit lamp biomicroscopy revealed a normal anterior segment, with a normal iris configuration and a normal pupil diameter, with exposed mild nuclear sclerosis of the lens to dilatation. The left eye revealed pseudoexfoliative material on the inferior and temporal parts of the anterior capsule and in the border of the iris (Figure 1), which displayed an inverse “volcano shape”, corresponding to a reverse pupillary block, together with mild nuclear sclerosis of the lens.

**Figure 2 diagnostics-15-00758-f002:**
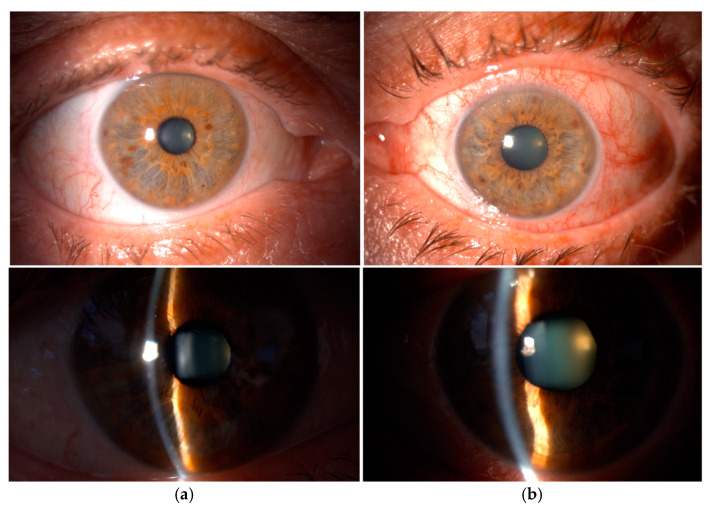
A notable difference in the anatomical configuration of the iris and anisocoria was found when comparing both eyes. The slit lamp biomicroscopy shows (**a**) a normal configuration of the anterior chamber and the iris in the right eye and (**b**) a narrow anterior chamber with a reverse pupillary block in the left eye with notable anisocoria between the eyes. Reverse pupillary block is a key mechanism in pigment dispersion syndrome (PDS) [1]. Increased anterior chamber pressure leads to backward bowing of the iris, creating irido-lenticular contact and a feedback loop that can heighten the pressure difference between the anterior and posterior chambers, leading to significant fluctuations in intraocular pressure (IOP).

**Figure 3 diagnostics-15-00758-f003:**
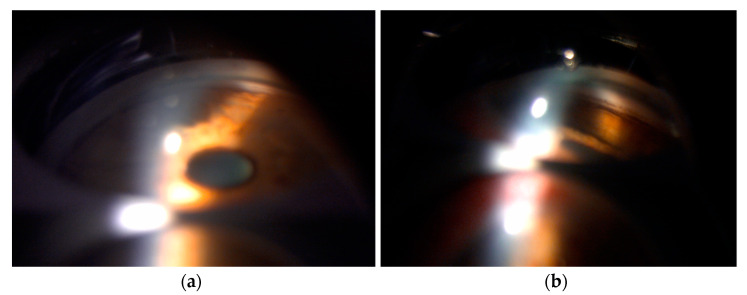
Gonioscopy and gonioscopic indentation showed notable differences between the two eyes regarding the condition of the angles (Figure 3). In the left eye, the examination revealed that the iris had a bulging appearance at the edge, followed by a significant dip or depression toward the pupil. This peripheral bulging suggests that there may be abnormal tissue or some other pathology underneath the iris, pushing it towards the anterior chamber and consequently closing the angle [2,3]. Additionally, an abundance of brown pigment was observed at and anterior to Schwalbe’s line. The accumulation of pigment in this area is consistent with the phenomenon known as the Sampaolesi line, which is characterized by a visible band of pigment extending along the peripheral iris. This finding is typically associated with PDS and PEX and reflects the dynamic interplay between the iris and the trabecular meshwork during ocular movements. The presence of such pigment can further obstruct aqueous outflow, exacerbating IOP elevation. In contrast, the right eye exhibited a markedly different profile during the gonioscopy examination. The angle was found to be open, allowing for unobstructed visibility of all the structures present within the anterior chamber, which suggests that this eye does not currently face the same risk of angle-closure complications as the left eye. The image in Figure 3 shows a gonioscopic indentation with (**a**) an open angle and normal iris configuration in the right eye and (**b**) a narrow angle with an initial bulging of the iris followed by a depression to the pupillary edge with the Sampaolesi line in the left eye.

**Figure 4 diagnostics-15-00758-f004:**
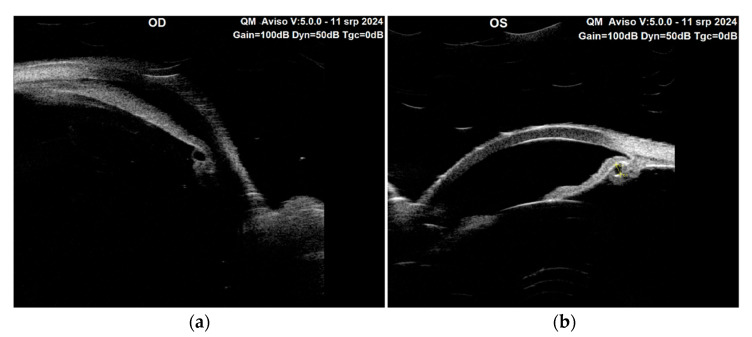
Ultrasound biomicroscopy (UBM) with 50 MHz transducer, (Quantel Aviso, Topcon, Tokyo, Japan) provided crucial insights into the structural characteristics of the anterior segment in both eyes, as illustrated in Figure 4. The UBM examination revealed the presence of cystic formations within and around the ciliary body in the left eye, which is often indicative of various pathologies including ciliary body cysts. Primary ciliary body cysts are benign, fluid-filled cavities from the iris and ciliary body, often found incidentally during eye examinations [4,5,6]. However, they can pose risks such as angle-closure glaucoma if they cause the anterior chamber angle to exceed 180 degrees or if multiple cysts displace the iris forward, resembling pseudoplateau iris [5]. Interestingly, cystic areas were also unexpectedly detected in the right eye, suggesting that there may be a bilateral component to the underlying pathology, despite the differences noted in angle status between the two eyes. In addition to the identification of these cysts, the UBM findings highlighted an axially wide anterior chamber in the left eye, which is typical of reverse pupillary block. This reduced chamber depth can be associated with several issues, including an increased risk of angle closure and elevated IOP. Furthermore, the examination confirmed the presence of a reverse pupillary block in the left eye. Figure 4 shows the performance of the UBM on the right eye (**a**) and the left eye (**b**) with ciliary body cysts pushing the peripheral iris and causing angle closure with an associated reverse pupillary block in the left eye. One possible bias in the execution of the UBM arises from the supine position in which it was conducted. This position may cause a posterior displacement of the lens, resulting in a deeper anterior chamber. It is crucial to take this positional change into account, as it could affect the interpretation of the images.

**Figure 5 diagnostics-15-00758-f005:**
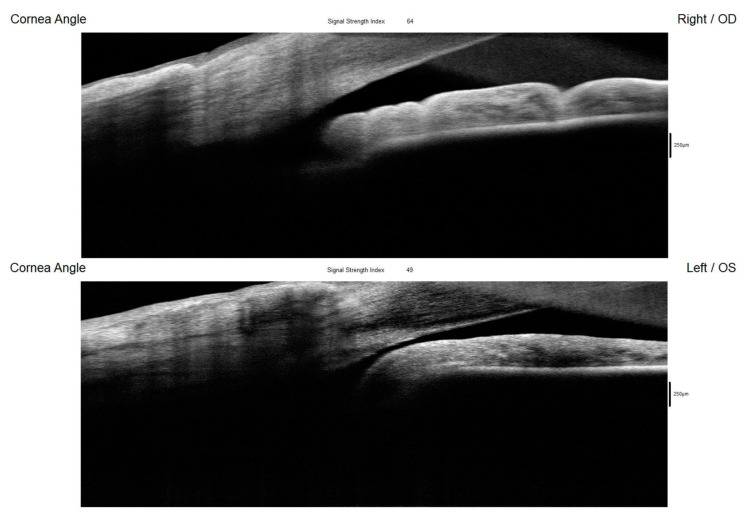
The AS-OCT (RTVue, Optovue Inc., Fremont, CA, USA) offered comprehensive imaging of the anterior segment structures, primarily emphasizing the cornea and corneal angle. While it also visualized the iris, its capability for retroiridian imaging was limited, preventing assessments of this space. The device revealed a narrow and closed angle in the left eye (Figure 5). This finding aligns with the results obtained from gonioscopy and UBM, which similarly indicated a compromised angle configuration. Figure 5 shows AS-OCT of the right and left eyes showing an open corneal angle in the right eye (upper picture) and a closed angle in the left eye (lower picture) resulting from the posterior displacement of the ciliary body cysts. OD: right eye; OS: left eye.

**Figure 6 diagnostics-15-00758-f006:**
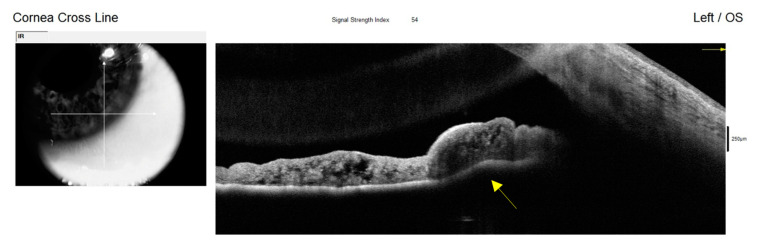
AS-OCT of the left eye showing a postoperative open corneal angle. The yellow arrow points to the presence of ciliary body cysts and the anterior displacement towards the anterior chamber angle. OS: left eye. The diagnosis of unilateral reverse pupillary block associated with ciliary body cysts and PEX was established, based on a combination of clinical findings, gonioscopy, UBM, and AS-OCT, which have emerged as invaluable diagnostic tools in these cases [7,8]. Evaluation of the anterior segment with Scheimpflug tomography (Pentacam; Oculus GmbH, Wetzlar, Germany) together with optical biometry (IOL Master 500, Carl Zeiss Meditec AG, Jena, Germany) was performed before and after surgery of the left eye to achieve a more objective evaluation of the anterior segment structures, including the anterior chamber depth (ACD), anterior chamber angle (ACA), lens thickness (LT), and axial length (AL), for a better understanding of their behaviour in this case. The comparison of eye measurements taken before and after surgery were taken for a better objective analysis. For the right eye (RE) before surgery, the anterior chamber depth (ACD) was 2.70 mm, the anterior chamber angle (ACA) was 29.8, the lens thickness (LT) was 4.78 mm, and the axial length (AL) was 23.04 mm. After the surgery, measurements from the left eye (LE) show an ACD of 2.66 mm, an ACA of 30.9, an LT of 4.76 mm, and an AL of 23.00 mm. The surgery was performed on the left eye (LE). These measurements helped in evaluating changes in the anterior chamber and lens thickness due to surgical intervention. While the right eye showed minimal changes, with slight decreases in the anterior chamber depth and axial length, the results highlight changes in the anterior segment parameters of the left eye, which was the only eye operated on. The anterior chamber depth increased notably from 1.90 mm to 3.23 mm, indicating a deepening likely due to the communication (trabeculectomy) made between the chambers to break the reverse pupillary block. Meanwhile, the anterior chamber angle decreased from 37.0° to 33.0°, suggesting some narrowing while remaining relatively wide. We believe that this may be due to the presence of the ciliary body cysts which are still causing pressure behind the iris in the angle. The lens thickness decreased from 5.11 mm to 4.78 mm, implying repositioning. These findings underscore the surgical impact on the left eye’s dynamics, essential for assessing postoperative outcomes. Posterior segment OCT revealed normal retinal nerve fibre layer and ganglion cell layer thicknesses in the right eye, along with an intact internal limiting membrane-to-retinal pigment epithelium thickness. In contrast, the left eye exhibited a more excavated neuroretinal rim compared with the right eye, and a reduced ganglion cell layer thickness, with an intact macular structure. Considering the clinical features observed in the patient, including significantly elevated IOP and noted impairment of the optic nerve and visual field, the decision was made to proceed with trabeculectomy with mitomycin C (MMC) in the left eye, rather than performing a peripheral iridotomy. In our evaluation during the slit lamp examination or while performing UBM, we did not observe any additional factors or unusual behaviours of the iris (iridodonosis) or the lens (phacodonosis). Additionally, during the surgery procedure (trabeculectomy), careful evaluation and measures were taken to see if there was any intraoperative reaction of the structures or the aqueous humour when the iridectomy was performed. Postoperative observations revealed control of the IOP. At one month and three months after the surgery, the IOP measurements demonstrated a significant improvement: the right eye measured 17 mmHg and 18 mmHg, while the left eye measured 10 mmHg and 11 mmHg, respectively. Importantly, these measurements were recorded without the use of any topical or systemic medications, indicating that the trabeculectomy successfully managed the IOP. Given the normal examination findings in the right eye and the absence of any clinical manifestations or concerns, it was decided to continue with a strategy of observation for the right eye, monitoring it without immediate intervention. This approach allows for ongoing assessment while minimizing unnecessary treatments. Overall, the outcomes of the trabeculectomy for the left eye were favourable, as evidenced by both the controlled IOP and the stability of the patient’s ocular health in the right eye. The way reverse pupillary block developed in our case is interesting. In a normal eye, aqueous humour is produced by the ciliary body and flows from the posterior chamber through the pupil into the anterior chamber, where it eventually drains out through the trabecular meshwork. In certain conditions, such as in the presence of ciliary tumours or iridociliary cysts, like in our case, we initially hypothesized that due to the anterior displacement that the iridociliary cysts make towards the anterior chamber angle, they can close the structure and lower the outflow of the aqueous humour. This can lead to an increase in pressure in the anterior chamber and consequently a misdirection of the aqueous humour to the posterior chamber. This change in aqueous humour dynamics between both chambers can cause the iris to bow backward towards the lens, creating a seal at the pupillary margin, and consequently predisposing it to the development of a pupillary block. As the iris moves backward (bowing), it may further exacerbate the blockage by pressing against the crystalline lens. As the pupillary aperture becomes more restricted due to the backward bowing of the iris and the resulting blockage, it becomes harder for aqueous humour to exit the posterior chamber. This creates a negative pressure effect, pulling the iris even further back. Therefore, the anatomical changes induced by the ciliary body cysts alter the normal dynamics of the iris and lead to increased contact with neighbouring structures, hindering aqueous humour outflow and leading to the development of a reverse pupillary block. Additionally, we believe that the pseudoexfoliative material, which consists of flaky, fibrillar deposits, along with any associated inflammatory components, may exacerbate this condition in a manner similar to the mechanisms observed in PDS [9]. In this syndrome, the shedding of pigment granules from the iris leads to increased IOP due to obstruction of the trabecular meshwork. In our hypothesis, the pseudoexfoliative material may similarly interact with the iris, resulting in inflammatory responses that further compromise the integrity of the anterior chamber angle. This interaction likely renders the iris more flaccid and susceptible to stretching and displacement, particularly in the presence of elevated pressures or abnormal anatomical configurations. As the iris becomes more pliable, it may lose its normal rigidity and structural support, allowing it to bow or shift more easily in response to mechanical forces, such as those exerted by the ciliary body cysts or the accumulation of pseudoexfoliative material. As a result of these changes, the iris might adhere to regions populated with pseudoexfoliative material within the anterior segment of the eye. This adherence can create a seal that prevents aqueous humour from flowing freely from the posterior to the anterior chamber, effectively leading to a reverse pupillary block. This reversed condition not only contributes to an additional elevation of IOP but may also facilitate the progression of glaucomatous changes, as the impaired drainage exacerbates the pressure buildup within the eye. Another potential hypothesis is that this patient may be experiencing both PDS and PEX concurrently, described in the literature as an overlap syndrome [10]. Although there is limited research on the simultaneous occurrence of these two conditions, some reported cases have indicated that individuals can exhibit both pathologies [11]. This phenomenon invites consideration of whether the simultaneous presence of these syndromes occurs by chance or is the result of the manifestation of two separate genetic mutations in the same individual. It is important to note that while PDS and PEX share certain clinical features, they are generally recognized as distinct clinical entities, each with their own genetics. For instance, while PDS is characterized by the shedding of pigment granules from the iris, PEX involves the accumulation of fibrillar material on the lens and other intraocular structures; both contribute to increased IOP and other ocular complications in different ways, as well as the risk of developing glaucoma. To the best of our knowledge, we present a very rare case of bilateral ciliary body cysts, manifesting unilaterally with reverse pupillary block, in conjunction with PEX, subsequently leading to angle-closure glaucoma. The interplay between ciliary body cysts and pseudoexfoliative material, coupled with associated inflammatory responses, plays a critical role in altering iris dynamics. This alteration can set off a cascade of events culminating in reverse pupillary block and its associated complications. The iridotrabecular contact induced by ciliary body cysts leads to an increase in aqueous flow resistance, elevating anterior chamber pressure that induces a backwards displacement of the iris, increasing iris–lens contact and reducing transpupillary aqueous humour flow. The anatomical changes of the iris in PEX could favor the appearance and further persistence of a reverse pupillary block.

## Data Availability

All relevant data are within the manuscript.

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
