# Peer review of "Unilateral Reverse Pupillary Block Associated with Multiple Ciliary Body Cysts and Pseudoexfoliative Syndrome"

_diagnostics, 2025, doi:10.3390/diagnostics15060758_

Round 1

Reviewer 1 Report

Comments and Suggestions for Authors

Dear Authors,

In the case report titled "Unilateral Reverse Pupillary Block (RPB) Associated with Multiple Ciliary Body Cysts and Pseudoexfoliative Syndrome", a PEX case developing Reverse pupillary block due to ciliary body cyst is presented and the importance of multimodeling is emphasized. My opinions regarding this case report are stated below.

  1. RPB is generally defined in pigment dispersion syndrome (PDS) and is the disruption of the pressure balance between the anterior and posterior chambers. The configuration of the iris is very important in diagnosis. The presence of pseudoexfoliation (PEX) in the case of the study is interesting, and the association of PDS and PEG has been reported in the literature as overlap syndrome. (Current Opinion in Ophthalmology 36(2):p 122-129, March 2025.)
  2. In this case, it is important to evaluate anterior segment with topography in multimodelling, and giving anterior chamber depth (ACD) and anterior chamber angle (ACA) width values ​​will provide objective evaluation.
  3. In addition, in this and similar cases, it is very important to evaluate the lens thickness and lens position. It would be appropriate to discussion.
  4. Narrow-angle glaucoma may also develop in PEX cases. In addition, in the PES, has a high risk of developing weakness in the zonules and resulting lens subluxations and pupillary block. Was there  iridodonosis or phacodonosis in this case?
  5. In UBM, the image was not pass from the same region in the right and left eye images.(Fig 4) Additionally, UBM examination in the supine position may cause the lens to shift backwards, causing the anterior chamber to become deep.
  6. What was the AC OCT appearance of the anterior segment image in Fig.2 b? This is valuable to show the iris configuration .
  7. It would be appropriate to provide information about the devices used in measurement.
  8. The treatment of this case should be stated in detail. In general, information should be given about the treatment approach in RPD and what was done for this case should be reported. 
  9. Is the area where ciliary body cysts push the iris forward sufficient for angle-closure glaucoma? (Fig. 6) It would be valuable to provide literature on this subject.

Response to Reviewer 1 Comments

1. Summary

2. Questions for General Evaluation

Reviewer’s Evaluation

Response and Revisions

Does the introduction provide sufficient background and include all relevant references?

Yes/Can be improved/Must be improved/Not applicable

In the initial version of the manuscript, several points raised by the reviewer were indeed addressed. However, in adapting the manuscript for a special issue, we needed to modify it to align with that specific section. We have added to the manuscript the most important points that the reviewer has highlighted to make the article more consistent.

Is the research design appropriate?

Yes/Can be improved/Must be improved/Not applicable

Are the methods adequately described?

Yes/Can be improved/Must be improved/Not applicable

Are the results clearly presented?

Yes/Can be improved/Must be improved/Not applicable

Are the conclusions supported by the results?

Yes/Can be improved/Must be improved/Not applicable

3. Point-by-point response to Comments and Suggestions for Authors

[Comment 1] RPB is generally defined in pigment dispersion syndrome (PDS) and is the disruption of the pressure balance between the anterior and posterior chambers. The configuration of the iris is very important in diagnosis. The presence of pseudoexfoliation (PEX) in the case of the study is interesting, and the association of PDS and PEG has been reported in the literature as overlap syndrome. (Current Opinion in Ophthalmology 36(2):p 122-129, March 2025.)

[Response 1] Thank you for sharing this insightful comment and for the terminological clarification that we can adapt to our case. The relationship between pigment dispersion syndrome (PDS) and pseudoexfoliation syndrome (PEX) is indeed both rare and interesting, with only one case reported/found in the literature.* In the context of the reverse pupillary block  (RPB) that our patient presented, we were surprised to clearly diagnose PEX, especially since PDS is usually the syndrome associated with RPB. That’s the reason why we propose a few theories about our findings, including the idea of an overlap syndrome, as you mentioned in your comment. But what’s really striking about the case is that it’s a confirmed PEX associated with an RPB.

*Pokrovskaya, O., & O’Brien, C. (2016). What’s in a gene? Pseudoexfoliation syndrome and pigment dispersion syndrome in the same patient. Case Reports in Ophthalmology, 7(1), 54–60. https://doi.org/10.1159/000443697

[Comment 2] In this case, it is important to evaluate anterior segment with topography in multimodelling, and giving anterior chamber depth (ACD) and anterior chamber angle (ACA) width values ​​will provide objective evaluation.

[Comment 3] In addition, in this and similar cases, it is very important to evaluate the lens thickness and lens position. It would be appropriate to discussion.

[Response 2,3] Thank you for your comment. We did and we'll incorporate this data into the case report to strengthen it. Updated in the manuscript.

Before surgery

RE

LE*

ACD

2.70 mm

1.90 mm

ACA

29.8º

37.0º

LT

4.78 mm

5.11 mm

AL

23.04 mm

23.00 mm

After surgery

RE

LE*

ACD

2.66 mm

3.23 mm

ACA

30.9º

33.0º

LT

4.82 mm

4.78 mm

AL

23.00 mm

23.00 mm

ACA Anterior chamber angle, ACD Anterior chamber depth, AL axial length, LT lens thickness, RE right eye, LE left eye.

*Eye where the surgery was performed

[Comment 4] Narrow-angle glaucoma may also develop in PEX cases. In addition, in the PES, has a high risk of developing weakness in the zonules and resulting lens subluxations and pupillary block. Was there  iridodonosis or phacodonosis in this case?

[Response 4] Thank you so much because this is a question that we considered as well while evaluating the case before planning our management. In our evaluation during the slit lamp examination or while performing the UBM, we did not observe any additional factors or unusual behaviors of the iris or the lens. Additionally, during the surgery procedure (trabeculectomy) we watched carefully to see if there was any intraoperative reaction of the structures or the aqueous humor when the iridectomy was performed. Everything appeared normal as well. Updated in the manuscript.

[Comment 5] In UBM, the image was not pass from the same region in the right and left eye images.(Fig 4) Additionally, UBM examination in the supine position may cause the lens to shift backwards, causing the anterior chamber to become deep.

[Response 5] Thank you for your observation. We acknowledge that the UBM images were taken from slightly different regions in the right and left eyes, which may affect direct comparisons. Additionally, conducting UBM in the supine position can indeed lead to a posterior shift of the lens, resulting in a deeper anterior chamber. This positional change is important to consider, as it may influence the interpretation of the images. We will make sure to address these factors in our analysis. Updated in the manuscript.

[Comment 6] What was the AC OCT appearance of the anterior segment image in Fig.2 b? This is valuable to show the iris configuration .

[Response 6] Our anterior segment OCT (RTVue, Optovue Inc., Fremont, CA) captures various anterior segment components, including the cornea, corneal mapping, corneal angle and iris but it does not capture a comprehensive image of the entire anterior chamber or full anterior segment. Instead, it primarily focuses on specific structures (mostly corneal structures and corneal angle) and can visualize the iris, although it has limited retroiridian visualization so it cannot provide information of the retroiridian space. Updated in the manuscript.

[Comment 7] It would be appropriate to provide information about the devices used in measurement.

[Response 7] Thank you very much for your observation. We will take this comment into account and provide more specific information regarding the devices used. Updated in the manuscript.

[Comment 8] The treatment of this case should be stated in detail. In general, information should be given about the treatment approach in RPD and what was done for this case should be reported.

[Response 8]  Thank you for your valuable comment. In the initial version of the manuscript, we included a paragraph on the treatment, management, and evolution of the patient. However, when we submitted the manuscript for the special issue "Interesting Images," the treatment section was not contemplated on the publication criteria. We will ensure it is added back in. Updated in the manuscript.

We leave you the paragraphs written in the first manuscript version with all the information, for your interest:

“Posterior segment OCT revealed normal retinal nerve fibre layer and ganglion cell layer thickness in the right eye, along with an intact internal limiting membrane to retinal pigment epithelium thickness. In contrast, the left eye exhibited a more excavated neuroretinal rim compared with the right eye, and reduced ganglion cell layer thickness, with an intact macular structure.

Standard automated perimetry by Swedish Interactive Thresholding Algorithm (SITA) testing demonstrated preserved field in the right eye and an absolute defect in the left eye.

After the initial examination, the patient's IOP was found to be poorly controlled, despite being treated with the maximum allowable topical and systemic ocular medications. Given this lack of improvement, surgical intervention for the left eye was deemed necessary. The diagnosis was established based on a combination of clinical findings, gonioscopy, UBM and AS-OCT which indicated a unilateral reverse pupillary block associated with ciliary body cysts and PEX.

Considering the clinical features observed in the patient, including significantly elevated IOP and noted impairment of the optic nerve and visual field, the decision was made to proceed with trabeculectomy with mitomycin C (MMC) in the left eye, rather than performing a peripheral iridotomy.

Postoperative observations revealed control of the IOP. At one month and three months after the surgery, the IOP measurements demonstrated significant improvement: the right eye measured 17 mmHg and 18 mmHg, while the left eye measured 10 mmHg and 11 mmHg, respectively. Importantly, these measurements were recorded without the use of any topical or systemic medications, indicating that the trabeculectomy successfully managed the IOP.

Given the normal examination findings in the right eye and the absence of any clinical manifestations or concerns, it was decided to continue with a strategy of observation for the right eye, monitoring it without immediate intervention. This approach allows for ongoing assessment while minimizing unnecessary treatments. Overall, the outcomes of the trabeculectomy for the left eye were favourable, as evidenced by both the controlled IOP and the stability of the patient's ocular health in the right eye.”

[Comment 9] Is the area where ciliary body cysts push the iris forward sufficient for angle-closure glaucoma? (Fig. 6) It would be valuable to provide literature on this subject.

[Response 9]  Thank you for your comment. The area where ciliary body cysts push the iris forward can indeed play a role in angle-closure glaucoma and there is a paragraph on the case in which we discuss this based in some literature but we had to shorten it and consequently remove it due to the section we were applying to (“Interesting Images”). We will take your suggestion into consideration and look to include literature that addresses this relationship in our discussion. Your feedback is greatly appreciated. Updated in the manuscript.

1)     Lliteras Cardin, M. E., Pacheco Várguez, J. A., Espinosa-Rebolledo, A. E., & Méndez-Domínguez, N. (2021). Angle-closure glaucoma secondary to ciliary body cysts treated with subliminal transscleral cyclophotocoagulation. Report of a case. Archivos de La Sociedad Española de Oftalmología (English Edition), 96(12), 653–657. https://doi.org/10.1016/j.oftale.2020.10.010

2)     https://glaucomatoday.com/articles/2016-nov-dec/a-case-of-ciliary-body-cysts

4. Response to Comments on the Quality of English Language

Point 1:

Response 1: The English is fine and does not require any improvement.

5. Additional clarifications

No applicable.

Reviewer 2 Report

Comments and Suggestions for Authors

I have seen innumerable cases of Pigment dispersion syndrome , PEX glaucoma and ciliary body cysts - one case has been explained in our book just recently but this case report is very different in presentation. I have still to find a case where these two conditions co exist along with iris cysts which seem to be the initiating factor in creating features of the two syndromes. The  hypothesis given for explaining the condition is good , plausible and understandable .Overall a good case report but it has missing information about surgery/ laser iridotomy and Antiglaucoma drugs. A picture of the optic nerve heads of the two eyes would have shown the damage due to glaucoma. The IOPs post op have not been stated

Author Response

Response to Reviewer 2 Comments

1. Summary

2. Questions for General Evaluation

Reviewer’s Evaluation

Response and Revisions

Does the introduction provide sufficient background and include all relevant references?

Yes/Can be improved/Must be improved/Not applicable

Thank you for your valuable comments and for your opinion. In the initial version of the manuscript, several points raised by the reviewer were indeed addressed. However, in adapting the manuscript for a special issue, we needed to modify it to align with that specific section. We have added to the manuscript the most important points that the reviewer has highlighted to make the article more consistent.

Is the research design appropriate?

Yes/Can be improved/Must be improved/Not applicable

Are the methods adequately described?

Yes/Can be improved/Must be improved/Not applicable

Are the results clearly presented?

Yes/Can be improved/Must be improved/Not applicable

Are the conclusions supported by the results?

Yes/Can be improved/Must be improved/Not applicable

3. Point-by-point response to Comments and Suggestions for Authors

[Comment 1] I have seen innumerable cases of Pigment dispersion syndrome , PEX glaucoma and ciliary body cysts - one case has been explained in our book just recently but this case report is very different in presentation. I have still to find a case where these two conditions co exist along with iris cysts which seem to be the initiating factor in creating features of the two syndromes. The  hypothesis given for explaining the condition is good , plausible and understandable . Overall a good case report but it has missing information about surgery/ laser iridotomy and Antiglaucoma drugs. A picture of the optic nerve heads of the two eyes would have shown the damage due to glaucoma. The IOPs post op have not been stated.

[Response 1] Thank you for your valuable comments and for your opinion. This is the first time that we see a case, involving multiple pathologies like this and we engaged in extensive discussions about the potential explanations for the clinical presentation and manifestations. The relationship between pigment dispersion syndrome (PDS) and pseudoexfoliation syndrome (PEX) is indeed both rare and interesting, and we found only one case reported in the literature.* In the context of the reverse pupillary block  (RPB) that our patient presented, we were surprised to clearly diagnose PEX, especially since PDS is usually the syndrome associated with RPB. That’s the reason why we propose a few theories about our findings, including the idea of an overlap syndrome, as you mentioned in your comment recently published in your book. But what’s really striking about the case is that it’s a confirmed PEX associated with an RPB, which is really rare. Could you provide the reference of your book? It would be really valuable. Updated in the manuscript.

*Pokrovskaya, O., & O’Brien, C. (2016). What’s in a gene? Pseudoexfoliation syndrome and pigment dispersion syndrome in the same patient. Case Reports in Ophthalmology, 7(1), 54–60. https://doi.org/10.1159/000443697

In reference to missing information about surgery/ laser iridotomy and Antiglaucoma drugs:

In the initial version of the manuscript, we included a paragraph on the treatment, management, and evolution of the patient. However, when we submitted the manuscript for the special issue "Interesting Images," the treatment section was not contemplated on the publication criteria. We will ensure it is added back in. Updated in the manuscript.

We leave you the paragraphs written in the first manuscript version with all the information, for your interest:

“Posterior segment OCT revealed normal retinal nerve fibre layer and ganglion cell layer thickness in the right eye, along with an intact internal limiting membrane to retinal pigment epithelium thickness. In contrast, the left eye exhibited a more excavated neuroretinal rim compared with the right eye, and reduced ganglion cell layer thickness, with an intact macular structure.

Standard automated perimetry by Swedish Interactive Thresholding Algorithm (SITA) testing demonstrated preserved field in the right eye and an absolute defect in the left eye.

After the initial examination, the patient's IOP was found to be poorly controlled, despite being treated with the maximum allowable topical and systemic ocular medications. Given this lack of improvement, surgical intervention for the left eye was deemed necessary. The diagnosis was established based on a combination of clinical findings, gonioscopy, UBM and AS-OCT which indicated a unilateral reverse pupillary block associated with ciliary body cysts and PEX.

Considering the clinical features observed in the patient, including significantly elevated IOP and noted impairment of the optic nerve and visual field, the decision was made to proceed with trabeculectomy with mitomycin C (MMC) in the left eye, rather than performing a peripheral iridotomy.

Postoperative observations revealed control of the IOP. At one month and three months after the surgery, the IOP measurements demonstrated significant improvement: the right eye measured 17 mmHg and 18 mmHg, while the left eye measured 10 mmHg and 11 mmHg, respectively. Importantly, these measurements were recorded without the use of any topical or systemic medications, indicating that the trabeculectomy successfully managed the IOP.

Given the normal examination findings in the right eye and the absence of any clinical manifestations or concerns, it was decided to continue with a strategy of observation for the right eye, monitoring it without immediate intervention. This approach allows for ongoing assessment while minimizing unnecessary treatments. Overall, the outcomes of the trabeculectomy for the left eye were favourable, as evidenced by both the controlled IOP and the stability of the patient's ocular health in the right eye.”

4. Response to Comments on the Quality of English Language

Point 1:

Response 1: The English is fine and does not require any improvement.

5. Additional clarifications

No applicable.

Round 2

Reviewer 1 Report

Comments and Suggestions for Authors

Dear Authors, It has been determined that you have made the requested revisions in the article titled "Unilateral Reverse Pupillary Block (RPB) Associated with Multiple Ciliary Body Cysts and Pseudoexfoliative Syndrome".

Thank you

Reviewer 2 Report

Comments and Suggestions for Authors

Thank you for further explaining the case. It is a very unusual case of an overlap syndrome. You may not have had optic nerve pictures to show the state of the optic nerve or do further investigations to show optic nerve damage.